# MightyU – A portable sensor-based video game application for exercise training of children and adolescents with cerebral palsy

Lynn Eitner[1]*, Lennart Lücke[1,2], Elnaz Farshadfar[3], Christian Grüneberg[3], Christoph Maier[1], Almut Weitkämper[1], Bettina Menzen[1], Anja Burmann[4], Roman von Gehlen[5], Peter Konrad[5], Thomas Immich[6], Britta Karn[6], Patrick Filipowicz[7], Maximilian Pilk[7], Thomas Lücke[1]

**1** Department of Neuropaediatrics, University Children's Hospital, Ruhr University Bochum, Bochum, Germany, **2** Department of Anaesthesiology and Intensive Care Medicine, Campus Charité Mitte und Charité Campus Virchow-Klinikum, Charité-Universitätsmedizin, Berlin, Germany, **3** Division of Physiotherapy, Bochum University of Applied Sciences, Germany, **4** Fraunhofer Institute for Software and Systems Engineering ISST, Dortmund, Germany, **5** Velamed GmbH Science in Motion, Medical Technology and Biomechanical Concepts, Cologne, Germany, **6** Centigrade GmbH Saarbrücken, Germany, **7** Meap GmbH Agency for Communication and Design, Witten, Germany

* Lynn.Eitner@ruhr-uni-bochum.de

## Abstract

### Introduction

To prevent therapy fatigue and maintain motivation for daily home muscle training is important for children with cerebral palsy (CP). Therefore, we developed the computer-based Motion-controlled training tool *MightyU.* Its feasibility, short-term effectiveness and acceptance of the game in daily muscle training at home was now tested in children with varying degrees of motor impairment.

### Methods

A surface electromyography sensor detects muscle activation, which is translated into in-game actions. In this way, targeted muscle activity is used to collect coins during gameplay. 19 children with CP tested *MightyU* at home for a week on a predetermined muscle group of the upper or lower limbs. The feasibility analysis considered the number of refusals to participate in the study, voluntary use at home and feedback based on the Game Experience Questionnaire (GEQ). The evaluation of usability based on modified System Usability Scale (SUS). The training effect was assessed by analyzing the difference between collected coins before and after a one-week training.

**Data availability statement:** All relevant data are within the manuscript and its Supporting Information files.

**Funding:** This study has been funded by the Federal Ministry of Education and Research (Bundesministerium für Bildung und Forschung [BMBF], trial registration 13GW0299D). The project consisted of different work packages, including personnel and material costs. Clinical scientific work packages: TL and LE. Technological and development work packages: AB, RvG, PK, TI, BK, PF and MP. Beside this there have been no involvement into the study. The funders had no role in study design, data collection and analysis, decision to publish, or preparation of the manuscript.

**Competing interests:** The authors have declared that no competing interests exist.

## Results

*MightyU* was refused by 2 of 21 children, 19 children (N = 9 female, 11.3 ± 2.9 years, gross motor function classification scale GMFCS I-IV) used it at home without adverse effects. All children and their families exhibited great interest in this game independent of age, intelligence quotient, severity of disability, targeted movement, and prior experience with computer games. Key results from the GEQ were positive, yet children evaluated the gaming experience more positively to their parents across all categories. Median SUS score was 83.3% (IQR: 75.0–91.7) for children and 79.2% (IQR: 66.7–91.7) for parents, indicating good perceived usability. Training led to improvement in collecting coins (41% increase).

## Conclusion

There is a fundamental interest amongst children with CP and their families for the pioneering therapy option *MightyU* due to its user satisfaction and usability, thereby potentially augmenting patient autonomy and compliance.

## Introduction

Cerebral palsy (CP) is a neurological disorder that presents with symptoms ranging from mild to severe [1]. Individuals with CP experience impairment in motor function, cognitive performance, sensory perception, and speech, resulting in a decreased level of independence and quality of life. The management of children with CP requires rigorous and comprehensive therapies to preserve and enhance the abilities of children with CP and support their functional autonomy. Hence, it is imperative that the planning of therapeutic interventions be multidisciplinary and tailored to the child's abilities to maximize activity and participation [2].

Daily and consistent implementation of muscle training is necessary for preserving and expanding muscular capabilities, thereby promoting autonomy and participation to the highest degree achievable [1]. Oftentimes, not only the children with CP themselves, but also their families participate in the rehabilitative therapies or associated studies [3]. While family participation might have a positive effect on the patients motivation, caregiving demands were found directly contribute to both the psychological and the physical health of the caregivers (mostly family members) [4].

One of the main challenges is to maintain necessary motivation for therapy over a long period of time [5]. Limited availability of interactive home-training devices leads to repetitive and sometimes monotonous routines, so a high level of patient compliance is required to achieve training success [6]. This indicates the need for innovative and fun training methods providing a low threshold for home-practice to the children.

Gamified training applications have shown great potential to enhance neurological motor skills, particularly in pediatric research, but the systems are not always suitable for home-therapy due to their high cost and lack of portability [7–9]. So-called

exergames aim to use physical exercises and targeted movements to operate a computer game, e.g., via motion sensors or motion capture.

In the context of CP, training with exergames has also been researched in many cases involving the use of commercially available sensor devices with motion capture (such as Nintendo Wii or Xbox Kinect) or using specifically developed hard- and software for some studies [10–12]. The disadvantage of the latter is that they do not specifically consider the individual motor deficits of the patient group. In addition, it remains to be seen whether movement recognition can be used for all types of movement disorders, as many studies have only included patients with mild impairments [10].

A promising alternative method is the use of surface EMG (sEMG) for motion detection. Sensors can be placed on any muscle group, depending on individual training needs. This enables localized detection of muscle contractions, which, in combination with appropriate real-time visualization, can be used as biofeedback for highly specific neurocognitive training. Biofeedback is a method in which unconscious functions of the body are measured and presented to the user visually or acoustically. The goal is to gain better control over these functions [13]. Despite the predicted potential of sEMG for neurorehabilitation training as early as 2020 [14], we identified only few studies that have explored its usability for home neurorehabilitation training for children with CP in conjunction with video gaming [15–18]. This underlines the necessity for a newly designed home-training application tailored for CP patients.

The aim of the *MightyU* project was to develop and evaluate a gamified application that facilitates [1] cost-effective, secure, and engaging home-based training for children and adolescents with CP, [2] area-specific usage contingent on the necessary muscle group, and [3] adaptation based on the specific muscle contraction abilities of each person.

This manuscript reviews on the one hand the development process of the newly designed *MightyU* application, the feasibility in daily home training setting and the usability for children with CP and their parents and on the other hand the measurability of training effect analyzing the difference between gaming-scores before and after a one-week training period.

## Methods

This study was approved by the Ethics Committee of the Faculty of Medicine, Ruhr University Bochum in July 2020 (Reg. No. 20–6881) and was conducted in accordance with the Declaration of Helsinki. Written informed consent was obtained from all participants of the study and their parents or legal guardians.

### Participants

Participants were recruited between 28th October 2021 and 30th June 2022. Children and adolescents (aged 6–18 years) who were being treated for CP at the Social Paediatric Centre of the Department of Paediatrics and Adolescent Medicine, Ruhr University Bochum, Germany, were included in the study. All who matched the inclusion criteria were identified from the CP consultation list by the physician and the treating physiotherapist and were offered participation in the study. They were contacted either by telephone or personally at an outpatient appointment. Children classified within Gross Motor Function Classification System (GMFCS) levels I to IV were included in the study. This range represents individuals from those who can walk without limitations (level I) to those with limited self-mobility who may require assistive technology (level IV). This stratification allowed the inclusion of participants with mild to moderately severe motor impairments while excluding those with profound mobility limitations (GMFCS V), as they were unlikely use the game effectively [19]. During the first tests it became obvious that also children and adolescents with a below-average IQ (>50) were able to understand and use the application very well. The inclusion criterion was therefore extended to the condition of being able to understand and answer the questions in the questionnaire.

Exclusion criteria were unwillingness to participate (child or guardian), inability to provide informed consent, or significant language barriers in the family that prevented understanding of instructions or study materials.

## MightyU game application

**Development process.** Prior to the start, the home-training application *MightyU*, evolved through interprofessional collaboration, incorporating iterative testing on healthy individuals. This process ensured the application's foundational structure was robust, adaptable, and user-friendly. Previous to the start of the development process, the project partners conducted a qualitative survey to evaluate the needs and interests of individuals with CP. Decisions regarding game setup, play design and the use of a flying fantasy fox as gaming creature were based on results from survey. The development process followed a structured inter-professional approach, comprising of information technology-engineers, medical doctors, nurses, physiotherapists, and health scientists. Each discipline brought unique insights and expertise. The information technology team focused on creating a user-friendly interface and incorporating sensor technologies to accurately capture and interpret muscle movement. Doctors provided clinical insight into the specific needs and challenges of people with CP and guided the customization of exercises within the game application. Nurses played a critical role in understanding patient compliance and monitoring progress. Engineers were instrumental in the seamless integration of hardware and software components. A series of iteration loops were implemented throughout the development process, with early prototypes tested on healthy individuals. These testing phases allowed continuous refinement of the game application to ensure its safety, efficacy, and usability. Feedback from healthy participants was instrumental in fine-tuning the game application's parameters and exercises to match the abilities and limitations of people with CP. For further technical information regarding the development process of *MightyU* refer to Meister and Burmann [20].

## The game

Initial interviews were conducted among children and adolescents with CP. The aim was to determine the general interest in a game-based therapy and to find out wishes and ideas about the type and character of the game.

A suitable game setting and story were then designed. It was a game of skill in which a flying mythical creature (player character) collected coins in the sky over a defined period and had to avoid thunderclouds in order not to risk losing coins.

Physicians and physiotherapists were consulted on the implementation of muscle control of the game character. In the end, the team decided to use a sensor (Ultium sensor, Noraxon, USA) attached to the skin over the muscle group to be trained, which measure the intensity of the muscle contraction via sEMG (Fig 1). sEMG adhesive electrode sensor was

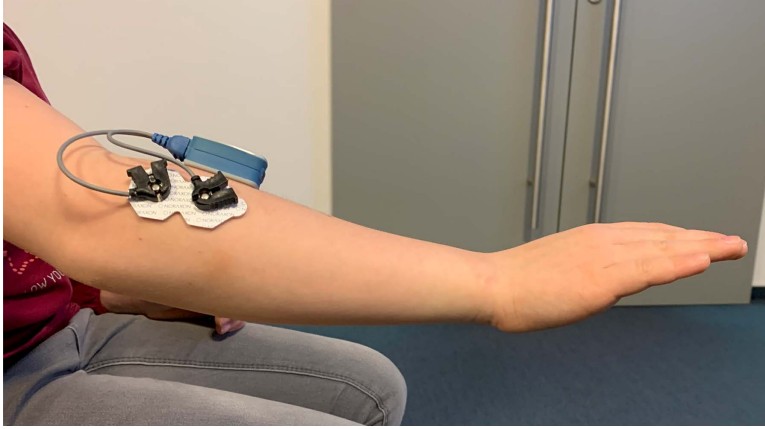

**Fig 1. sEMG sensor's location on a forearm (Ultium EMG, Noraxon, USA).**

applied to the respective muscle belly. The Bluetooth transponder was placed directly next to it and connected to the electrode via a cable. sEMG signals were A/D converted at 2000 Hz and band-pass filtered between 10 and 500 Hz. Subsequently, they were rectified and smoothed using a root mean square (RMS) with a 100 ms time window. Finally, the RMS signal was downsampled and transmitted via Bluetooth at 100 Hz. The transponder transmitted the sEMG signals to the game continuously, thus enabling control of the game character. The sampling rate for the game was 32 Hz. The values received in Unity are directly read from the microcontrollers; they were not further processed, only rounded.

When the muscle contracted, the player character rose from the bottom edge of the screen; when the muscle relaxed, it sank back to the bottom edge of the screen. Depending on the extend of muscle contraction (amplitude of the sEMG signal), the character's position could be influenced. This approach offers an advantage for home neurorehabilitation training and, to our knowledge, has not yet been used for this purpose.

**Muscle selection and sEMG sensor placement.** Muscle group selection was individualized and performed by experienced pediatric physiotherapists. It based on the child's specific motor deficits and the location of paresis (upper or lower limb). Based on these criteria, one movement was defined that was to be improved individually, e.g., dorsiflexion of the foot or wrist of the affected bodyside. The muscle contraction corresponding to this movement should be present at the beginning so that the game could be controlled and there was a certain chance of success for each child.

The sEMG sensors (Ultium EMG, Noraxon, USA) were placed over predefined anatomical locations corresponding to the selected muscles. Sensor positioning followed SENIAM standard surface EMG application guidelines [21]. It is typically attached at the muscle belly between origin and insertion points to optimize signal quality and reduce cross talk. Sensor placement was confirmed visually and via test contractions prior to gameplay. For the correct use at home, parents and children were given a detailed introduction on how to localize the area where the sensor should be stuck. Some families took photos, others marked the position with a pen. This one fixed position of the sensor was not changed over the week of training.

A brief calibration phase at the beginning of the training session was required to establish each child's maximum voluntary contraction and resting baseline of the muscle to be trained. This ensured that the character's reaction was adapted to the individual capacity of the muscle. Additionally, it was ensured that adjustments can be made depending on training progress or changes in the localization of the sEMG sensor. When used for training, the system should be calibrated before each training session to achieve training conditions that are adapted to the child's daily condition.

The player could choose between 3 levels of difficulty. On the easiest level, the player had to collect as many coins as possible, which appeared in different positions on the screen. There were only few occasional storm clouds as obstacles to avoid. The more coins collected, the faster they flew towards the character. At the medium difficulty level, the intervals between the storm clouds were shorter and there was a thundercloud field consisting of many clouds at different positions on the screen (Fig 2). The highest difficulty level had a 'grey area' at the top and bottom of the screen, which could be crossed by the character, but this resulted in the loss of coins. Therefore, children had to avoid maximum relaxation and contraction of the muscles.

In addition to the different levels described above for training purposes, there was one standardized level only to be used for study visits to create comparable conditions for all children included in the study. This level lasted 60 seconds, contained 20 coins to be collected, ran on a fixed gameplay speed and included no obstacles.

**Web portal.** Each participant's sEMG data was sent to a password-protected web portal at the end of a game session. The user's training history (number of training sessions, duration, sEMG maxima and minima and training success by number of coins) could be viewed and monitored under the user's ID, which was also available as raw data. The portal was designed that each participant could only access their personal training records and results. Therapists could view data of their own patients and study management had full data access but blinded using study ID's.

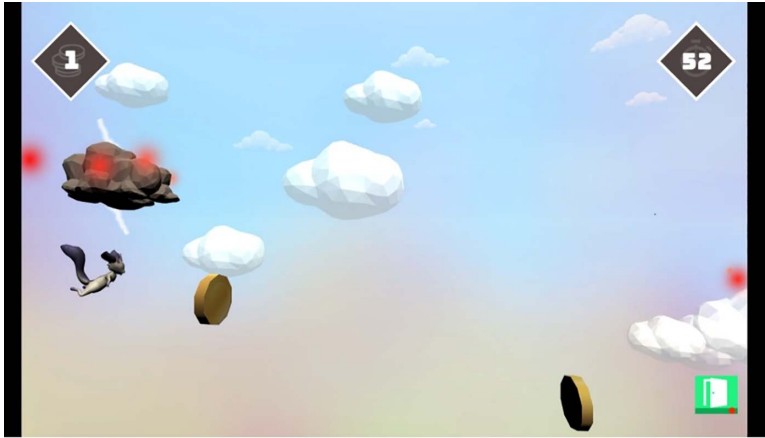

**Fig 2. Game screen with flying player character and coins to be collected.** Gameplay vision with mythical creature collecting moving objects (coins) and avoiding obstacles (storm clouds). Left upper edge: number of collected coins, right upper edge: time left until level ends in seconds, right lower edge: exit to leave the level.

## Assessment

To assess the feasibility and usability of the *MightyU* game application, we employed two established questionnaires tailored to the needs of children and adolescents with chronic illnesses, optimizing the evaluation process.

**Feasibility.** To measure gaming experience, we used the Gaming Experience Questionnaire (GEQ) [22–24] which is one of the most widely used instruments in gaming research. The original GEQ has 42 items. All questions in the test can be divided into 7 subscales covering different components of the gaming experience (challenge, competence, flow, immersion, tension, and positive and negative affect). We chose two questions from the GEQ, and three questions for negative affect, that can be assigned to the six experience dimensions. We did not include any questions from the tension domain (Supporting Material).

For each question there was a scale with 5 response options, from 'not at all' to 'extraordinarily'.

The scores for each component are calculated as the average of the individual questions [25]. Additional items were added to the questions to discuss adverse effects. ("I got dizzy or nauseous, I often didn't know what to do, I had muscle/joint pain while playing.")

**Usability.** To assess general usability, children and parents were asked three questions from the System Usability Scale (SUS), which is an established method in user research [26,27]. [1] I think I would like to use the system often. [2] I found the system to be unnecessarily complex. [3] I found the system easy to use. It was possible to choose an answer from a scale of five items, from 'strongly disagree' to 'strongly agree'.

**Training effect.** Quantification of training effect was assessed by comparing the number of collected coins before and after the one-week training period. sEMG data were reviewed by therapists but are not analyzed in this manuscript.

## Study visits procedure

During the first study visit, participants received a comprehensive briefing on the application and its technical aspects.

Following sensor attachment and calibration, participants played a standardized level lasting 60 seconds in a game-naive state, providing consistent baseline measurements. Subsequent supervised training sessions occurred at the clinic, followed by home-based sessions lasting up to 30 minutes, supervised by parents. Participants were able to decide for themselves how often they played during the 7-day period.

The second study visit conducted one week later, involved all participants sharing their experiences, replaying the standardized level, and completing relevant questionnaires.

After the first calibration of the system, no further calibration was performed during the one-week training phase and the second measurement. This served to create comparable examination conditions and thus also to compare the sEMG data before and after training.

## Statistical analysis

All numerical variables are presented as mean with standard deviation for the normal distribution, otherwise as median with interquartile range (IQR). The association between numerical parameters was tested for significance using Spearman's rank correlation test, differences in mean values were tested using the two-sided Mann-Whitney test, and associated parameters were tested for significance using the Wilcoxon test (significance level: $p < 0.05$). For this exploratory study, in which no hypotheses were tested, no corrections were made for multiple testing [28]. Microsoft Excel (365 Office, USA) and SPSS (Version 29.3, USA) served as the statistical tools for data analysis.

## Results

### Participants

21 children were contacted for participation, only 2 declined to participate due to following reasons: lack of interest in computer games (N = 1) and intolerance to any type of adhesives, electrodes, or sensors on the body (N = 1).

The final study group consisted of 19 children with considerable diversity in terms of age, physical limitation, and IQ (Table 1). Three children were unable to attend the second visit due to scheduling reasons, while 16 children finished the study and took part in both study visits. Only their data were analyzed to evaluate the game's efficacy and acceptance. Of these 16 children, 11 were able to walk without aids and 5 used a wheelchair for mobility. Overall, the system was used for one week of home-based training, ranging from 1 to 7 days (mean 3.9 ± 1.8 days). Nine participants (56%) engaged in playing *MigthyU* using their upper limb, mostly by extending their wrist joint (radiocarpal extension), while seven (44%) used their lower limb by extending their feet or knee joint. Children with CP GMFCS Level IV trained exclusively with the upper extremity, which was less severely impaired, to enable gaming success. Table 2 provides an overview of the participants' targeted muscle groups and training movements.

### Feasibility

Children and parents answered the GEQ questions similarly without significant differences. Parent's reports were made based on their observation.

Both figures displayed very high approval ratings and low negative ratings (Fig 3). No adverse effects were identified over the total study period. No episodes of vertigo, disorientation or nausea during the game (100% of children, 94% of parents answered "Not at all") were reported. 6% (N = 1) of parents reported that their child had muscle/joint pain while playing. The child did not confirm this in the questionnaire. 86% (N = 13) of children had either "exceptionally" or "very much" fun while playing. Parents reported similar results. Two children reported experiencing less moderate enjoyment. Three children discovered the game to be too arduous, yet the majority classified the game as challenging. The game's flow, which indicates the degree of engrossment, was deemed moderate as the game evidently did not captivate them to the point of forgetting their environment.

When grouped by experience dimensions, results were positive while no significant difference between parents and children's scores was observed (Fig 4). Median positive affect scores were high 3.5 and negative affect scores were low 0.3 among both children and parents, respectively. Median scores for flow 2, 2.5 and challenge 2, 1.3 indicate potential for improvement among both children and parents, respectively (Table 3).

**Table 1. Participants' clinical characteristics.**

| Clinical data | Intention to participate<br>n = 19 | Patients participated<br>n = 16 |
|---|---|---|
| female, n (%) | 9 (48) | 7 (45) |
| male, n (%) | 10 (52) | 9 (55) |
| age in years, mean | 11,2±3 | 11,1±2,9 |
| age in years, range | 6-17 | 6-16 |
| BMI, mean | 17,8±3,9 | 17,1±2,8 |
| **GMFCS, n (%)** | | |
| I | 7 (36) | 7 (45) |
| II | 6 (32) | 4 (25) |
| III | 3 (16) | 2 (13) |
| IV | 3 (16) | 3 (17) |
| **Paresis, n (%)** | | |
| hemiparesis | 9 (48) | 7 (45) |
| diparesis | 7 (36) | 6 (38) |
| triparesis | 1 (5) | 1 (7) |
| **Intelligence quotient** | | |
| range | 50-95 | 52-95 |
| age apropriate (85–115), n (%) | 7 (36) | 7 (45) |
| below average (70–84), n (%) | 7 (36) | 5 (32) |
| mild intellectual disability (50–69), n (%) | 5 (27) | 4 (25) |
| **Visus, n (%)** | | |
| visus reduction | 9 (43) | 8 (50) |
| strabismus convergens | 5 (27) | 4 (25) |
| **Hearing, n (%)** | | |
| no reduction | 19 (100) | 16 (100) |
| **Other diagnosis, n (%)** | | |
| epilepsy | 4 (19) | 1 (7) |
| hypothyreosis | 1 (5) | 1 (7) |
| small growth | 2 (10) | 2 (13) |
| **Treatment approaches, n (%)** | | |
| physiotherapy | 16 (85) | 13 (82) |
| occupational therapy | 10 (52) | 8 (50) |
| speech therapy | 5 (27) | 4 (25) |
| equine-assisted therapy | 3 (16) | 3 (17) |
| early intervention | 1 (5) | 1 (7) |

Patients who intended to participate in the study and those who finally participated.

## Usability

Children and parents answered the SUS-Survey highly positive. Minor differences between the two groups were not statistically significant (Table 4). Children and parents rated the game with a median of 83.3% and 79.2% of maximum achievable points, respectively (Table 4). Out of the 16 children (13 parents), 14 agreed that the game was easy to use. Only one child found it to be too complex, but still easy to use. 11 children (68%) expressed their desire to use the game more often.

**Table 2. Number of upper and lower extremity muscles examined.**

| Muscles | Movement | No. of patients |
|---|---|---|
| **Upper limb** | | 9 |
| Biceps brachii | Flexion of the forearm | 1 |
| Brachioradialis | Flexion of the forearm/ assists supination | 2 |
| Flexor digitorum profundus | Flexion of the fingers | 3 |
| Extensor carpi radialis longus | Extension of the wrist | 1 |
| Extensor digitorum | Extension of the fingers | 2 |
| **lower limb** | | 7 |
| Tibialis anterior | Dorsiflexion of the foot | 5 |
| Quadriceps femoris | Extension of the knee joint | 1 |
| Peroneus longus | Eversion oft the foot | 1 |

Target movement and muscle groups were determined by the physiotherapists. Each patient trains only one muscle group/movement.

## Training effect

During the week of training, almost all participants (68%) improved their performance, largely independently of the starting level. At a mean difference, children improved their total gaming scores between first and second assessment by 20% (Fig 5A). Exceptions were 3 children all with a GMFCS level IV and poor initial scores (Fig 5B, Purple lines).

It was observed that the target movements for most children were performed more precisely after training, i.e., in the post-measurement, than in the pre-measurement. This was partly reflected in the sEMG data over time. Fig 6 is an example of one child. Muscle contractions were less excessive and better controlled to reach the coins, which slid across the screen at different heights. Thus, in the pre-measurements, stronger contractions were performed more frequently than necessary. In the post-measurements, the control of force was improved in many cases.

## Discussion

The *MightyU* application was developed with the aim of providing a game that is fun to play, easy to use and both highly applicable and accessible for neurorehabilitative home training of children with CP. Training motivation, frequency and the proportion of self-training should be improved. The aim is to strengthen autonomy by maintaining and possibly even improving mobility. Four key findings were achieved with this pilot study. [1] The application was successfully developed based on a novel approach of recording the user's muscle potentials and kinematic data using a body worn sEMG sensor. The sensors detect the intensity of muscle activation, which in turn determines the extent of the character's movement. [2] The training application met with great interest among children and adolescents with CP. Only 2 of the 21 patients refused to participate. Feasibility analysis and gaming experience from the GEQ questionnaire of the *MightyU* application were positive. All participants voluntarily used the technology in a home setting several times during the training week without any adverse effects. [3] The usability of the system was also rated as good by 69% of parents and 75% of children and young people in the questionnaire. [4] A brief assessment of the measurability of the training effect showed an overall improvement in coin collection performance in 80% of the patients.

Virtual reality/ video gaming applications have been shown to be a potent tool for neurorehabilitative therapy. While several methods can be used to introduce gamification into therapeutic processes, including robotics-based systems, e.g., exoskeletons, controlling via balance boards, or motion capture via cameras, market-available devices are oftentimes primarily designed and calibrated for adults [29].

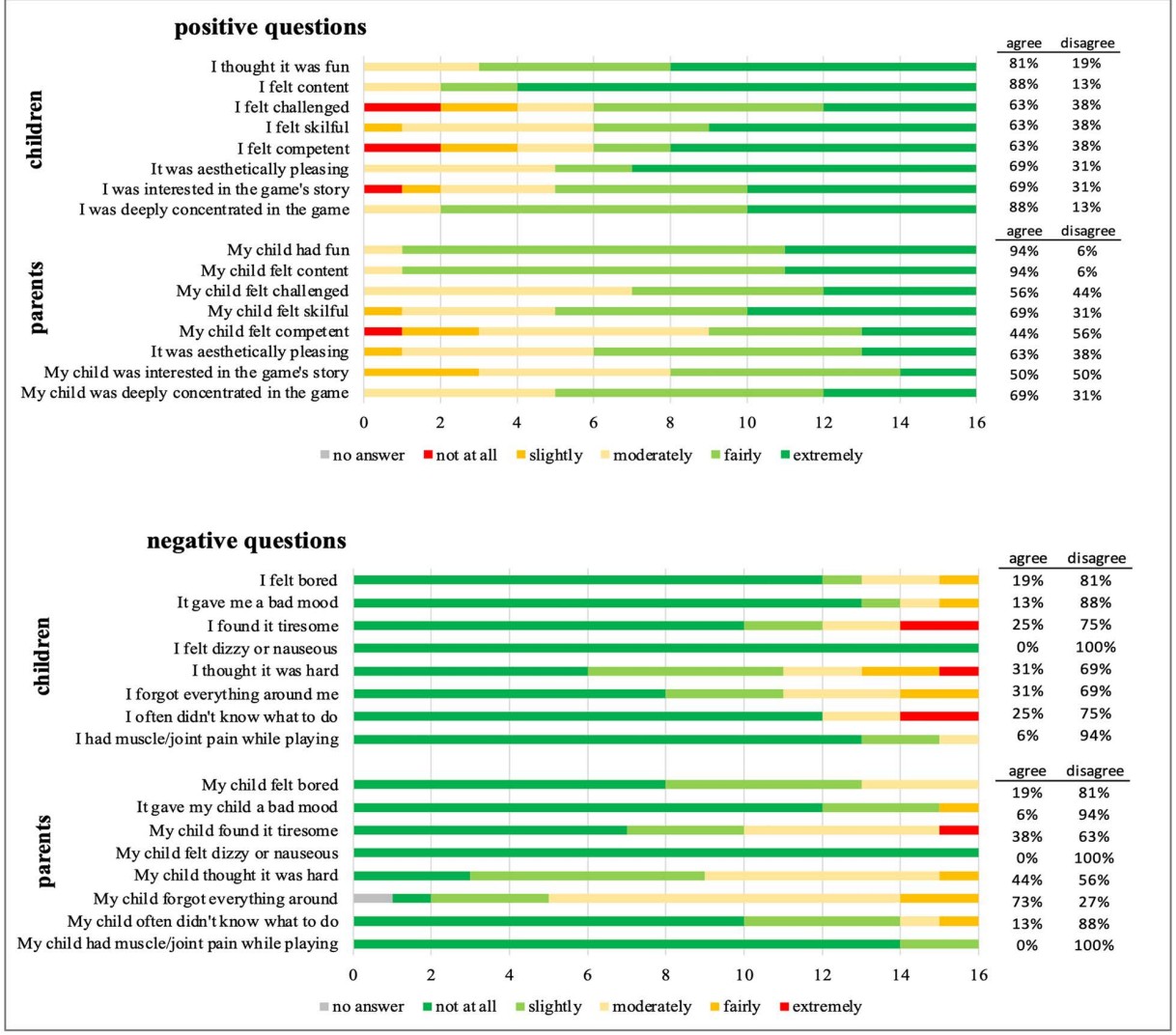

**Fig 3. Results from single questions of Game Experience Questionnaire by children and parents.** Stacked bar charts show responses to individual positively and negatively framed items of the GEQ. 16 children and their parents rated each item on a 5-point scale from 0 ("not at all applicable") to 5 ("extremely applicable"). For positive questions, colors range from dark green (strong agreement) to red (low agreement); for negative questions, the color scheme is reversed.

In their reviews Burin-Chu et al. and Kilcioglu et al. and found positive efficacy of commercially available VR tools for children with CP [8,10]. Similar positive results were identified in a review by Chen et al. [11]. However, the sizes of the included studies appear limited with only two of the 19 RCTs contained more than 20 individuals. When analyzing the training range of these market available devices, usability is limited to a broad training focus, such as training of the entire lower or upper extremity [30–33]. Such equipment does not allow for individualized training based on specific motor deficits, nor does it ensure home use due to limited availability and often high cost.

Interestingly, Chen et al. included in their review the results of a study analyzing the effectiveness of a special engineer-built training device. It showed a significantly larger training effect size, compared to other commercially

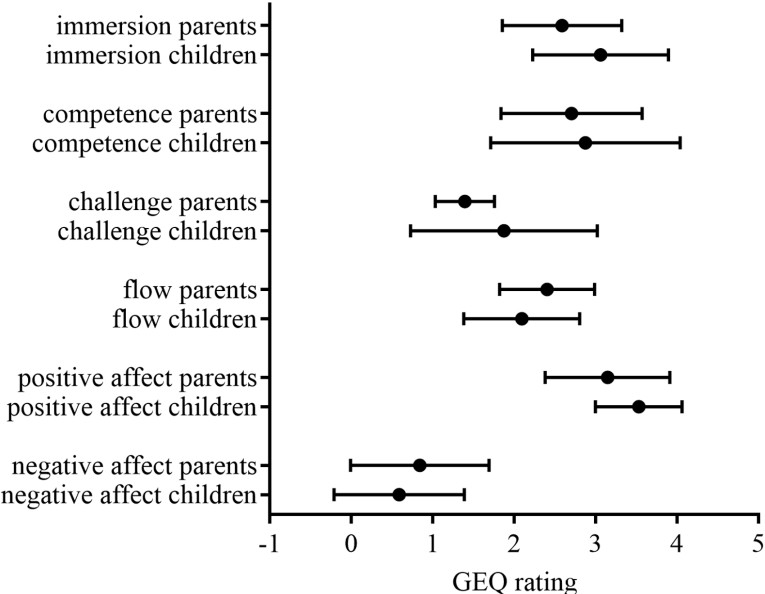

**Fig 4. GEQ Questionnaire results by parents and children according to GMFCS I-IV.** Mean scores (bullets) and standard deviations (whiskers) are shown for each experience dimension, rated on a scale from 0 ("not at all applicable") to 5 ("extremely applicable"). No significant differences between children's and parents' answers (p<0,05).

**Table 3. Game Experience Questionnaire results by parents and children.**

|  |  | Immersion | Competence | Flow | Negative affect | Positive affect | Challenge |
|---|---|---|---|---|---|---|---|
| Children | Median (Q25%; Q75%) | 3 (2.1; 3.9) | 3 (1.5; 4) | 2 (1.5; 2.5) | 0.3 (0; 1.4) | 3.5 (3; 4) | 2 (1; 3) |
| Parents | Median (Q25%; Q75%) | 2.5 (2; 3.4) | 2.5 (2; 3.5) | 2.5 (2; 3) | 0.7 (0.1; 1) | 3 (3; 4) | 1.3 (1.3; 1.7) |
| Wilcoxon U-test | p | 0.11 | 0.98 | 0.40 | 0.76 | 0.32 | 0.19 |
| correlation coefficient | R | −0.05 | 0.203 | 0.223 | 0.417 | 0.33 | 0.273 |

Questions grouped by experience dimensions, rated on a scale from 0 ("not at all applicable") to 5 ("extremely applicable"), median and 25% (Q25) and 75% (Q75) interquartile.

**Table 4. System usability scale results.**

|  |  | I think I would like to use the system often. | I found the system to be unnecessarily complex. | I found the system easy to use. | % of maximal feedback points |
|---|---|---|---|---|---|
| Children | Median (Q25%; Q75%) | 4.0 (3.0;5.0) | 1.0 (1.0;2.0) | 5.0 (4.0; 5.0) | 83.3% (75.0; 91.7) |
| parents | median (Q25%; Q75%) | 4.5 (3.0; 5.0) | 2.0 (1.0; 3.0) | 5.0 (4.0; 5.0) | 79.2% (66.7; 91,7) |
| Wilcoxon U-test | p | 0.72 | 0.18 | 0.48 |  |
| Correlation coefficient | R | 0.209 | 0.173 | 0.668 |  |

Responses based on a five-item scale, ranging from 1 ("strongly disagree") to 5 ('strongly agree"). The percentage of maximal feedback points reflects the proportion of the most positive potential answers (100%). Values are reported as medians with interquartile ranges (Q25–Q75).

available tools [11]. These findings are consistent with challenge point theory, which states that optimal task learning conditions require highly specific task training, such as that provided by engineer-built training applications [34].

This statement is even more important when considering the heterogeneous group of CP patients. This particular group of children presents some additional challenges that need to be considered for successful neurorehabilitative training.

| coins collected (%) | | | |
|---|---|---|---|
| patient | 1st assessment | 2nd assessment | delta |
| upper extremity | | | |
| #1 | 32% | 74% | 42% |
| #2 | 42% | 37% | -5% |
| #3 | 32% | 68% | 37% |
| #4 | 26% | 58% | 32% |
| #5 | 58% | 32% | -26% |
| #6 | 26% | 26% | 0% |
| #7 | 74% | 84% | 11% |
| #8 | 32% | 68% | 37% |
| #9 | 42% | 74% | 32% |
| lower extremity | | | |
| #10 | 89% | 100% | 11% |
| #11 | 95% | 100% | 5% |
| #12 | 53% | 79% | 26% |
| #13 | 26% | 53% | 26% |
| #14 | 58% | 95% | 37% |
| #15 | 68% | 89% | 21% |
| #16 | 32% | 63% | 32% |
| total | 49% | 69% | 20% |

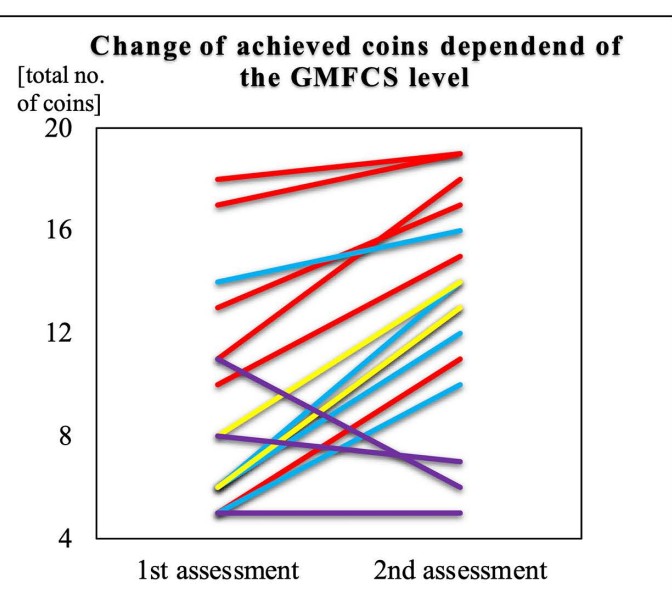

**Fig 5. Individual coin collection scores between first and second assessment.** A: percentage (%) of total achievable points collected per participant. B: stratified by GMFCS level. (GMFCS level I = red, level II = blue, level III = yellow, level IV = purple).

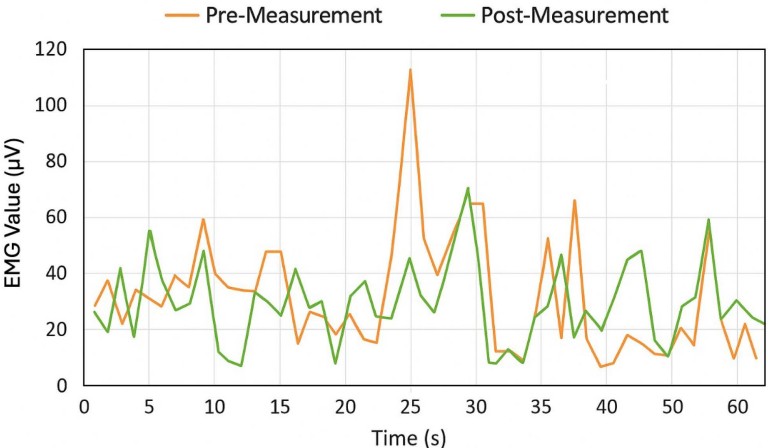

**Fig 6. Surface electromyography (sEMG) signals during standardized gameplay before and after training.** Example of a participant's sEMG (µV per second) recordings from biceps brachii over 60 seconds of gameplay (orange = pre-training, green = post-training (after one week).

Movement impairments such as paresis and spasticity, neurological limitations and cognitive impairments can occur individually or in combination and to varying degrees. Therefore, individualization of training goals and localized muscle groups is required. Avoidance of distraction or fatigue during play is necessary. The game setting must be easy to use and yet fun to play to maintain the motivation for therapy over a longer period of time. Finally, parents need to be enthusiastic about the technology as it requires them to be constantly present and involved in their children's therapy.

These specific requirements can only be met to a limited extent by the commercial tools available on the market, indicating the need for a more individualized approach.

In contrast to the previously mentioned approaches using commercial video gaming tools, the combination of video game with a specially developed sEMG technology [18] allows direct detection of even smallest muscle activations. Sensors can be placed freely based on individual deficits, enabling the most specific neurocognitive training. While Campanini et al. in 2020 formulated the great potential of sEMG for neurorehabilitation training [14], few studies have evaluated its usability in combination with video games in CP patients. We identified only two studies that evaluated the feasibility of sensor-based video training for neurorehabilitation:

Ona et al. conducted a feasibility analysis of a wrist-worn sEMG device in multiple sclerosis. VR-based training was performed over a period of 8 weeks, and although not statistically significant, clinical improvement in hand grip strength was demonstrated [35].

MacIntosh et al. conducted a feasibility analysis of a non-blinded study. 18 children with CP were enrolled. During the study period of 4 weeks, the training was done at home. While no adverse effects were reported, the study showed a moderate positive effect on measures of body function such as wrist extension and grip strength [17]. Both studies used a "MYO armband" for sEMG recording allowing home-training.

What made the *MightyU* project different from previous studies using standard commercial products was that both the technology and the game itself were newly developed and tailored to the needs of the target group. While the MYO armband's motion detection is limited to the execution of wrist gestures, the sensors used in our project allowed the intensity of muscle contraction to be quantified.

The requirements for the new applications were clearly defined before starting the development process to enable: [1] isolated induction of agonistic and antagonistic muscles to simulate physiotherapeutic training; [2] allow individual placement of sensors for muscle group specific contraction training; [3] allow individual adjustment of muscle contraction intensity during training based on direct feedback from sensors.

From the beginning of the project, a multidisciplinary team consisting of affected children, parents, nurses, physiotherapists, engineers, and doctors was involved in the planning, development and optimization of the game application.

Despite a heterogenous group with varying levels of GMFCS and locations of the paresis, participation rate remained high (16/19, Table 1) across study group (Fig. 5). No adverse effects were identified over the total study period. Positive results from GEQ and absence of adverse effects indicate feasibility for home training among the highly specific study group. These findings highly correlate with previous studies using sEMG for neurorehabilitation training [17,35].

Usability and enjoyment ranked highly positive with over 68% of tested children expressing their desire to use the game more often. While different feedback scales were used, MacIntosh at al. too, identified high satisfaction, motivation, and interaction over full study period [17]. Ona et al. used Client satisfaction questionnaire (CSQ-8), where median satisfaction of $80.35 \pm 10.93$ out of 100 points was even slightly higher when compared to our results [35].

In our study, game scores did not significantly improve between training days, but the overall training effect was positive (Fig 5). Interestingly, these observations are also consistent with the previously mentioned literature. While both studies presented did not find a significant improvement in gaming skills, the overall difference between pre- and post-intervention scores was positive [17,35]. Similar results were found in our study. Here, the subgroup analysis suggests that overall positive training effects can be observed for GMFCS levels I-III. The only exception were severely affected children with CP (GMFCS levels IV and V), for whom no improvement in playing ability was observed. The form of therapy may not be optimal for this classification. Even the lowest speed level of the coins still seems to be too fast to initiate the movement. However, our analysis of training effects is based on preliminary results with a small number of participants and would need to be verified in a larger study.

There are various rehabilitation games and programs, many of which are only available in clinics. The designed *MightyU* application allows a patient-centred approach to home practice by increasing autonomy and improving the

availability of training time. As a positive side-effect, this could support parents during play time. During our study, parents were required to continuously observe the children. In the future, it is conceivable that children will use the training application without parental supervision.

The exact extent to which improvement in game scores correlates with improvement in muscle, mobility and patient autonomy remains to be determined. In addition, analysis of the videographic scores allows precise detection of changes in mobility, providing insight into the effectiveness of *MightyU* training. Analysis of angular acceleration during movement provides detailed insight into dynamic motor function improvements. Assessment of baseline muscle tension before training, together with assessment of maximal muscle strength, provides valuable data on neuromuscular adaptations. Further elaboration of these analyses will be provided in subsequent papers.

A major challenge is the heterogeneous target group of children with CP. However, we interpret the heterogeneity of the subjects in our pilot study as positive, as a large spectrum of CP disease is represented and analyzed.

The present study has some limitations that should be considered when interpreting the results. Firstly, the number of cases in this study is small. However, the numbers are comparable to other studies in this population [11]. To strengthen the validity and generalizability of the results obtained, it would be necessary to replicate them in a larger collective. In addition, the study duration was only one week (1–7 days; mean 3.9 ± 1.8 days). However, a noticeable training effect on the muscles or mobility is not to be expected after such a short training session.

The *MightyU* application allows for customization of game requirements and difficulty levels. During the feasibility study, customization was omitted to gain a first understanding of the different skill levels. As a result, four subjects collected more than 70% of the maximum possible coins in the first evaluation, leaving limited room for improvement. Future game design should allow for a more defined adjustment of game difficulty and more game levels with varying degrees of difficulty to challenge players who are successful right from the start. All three subjects with GMFCS level IV showed no improvement in the more detailed assessment or support group. It is possible that a longer period of training may be required in this more severely impaired group to improve gaming skills.

Further research is needed to determine whether a longer training interval results in a stronger training effect and whether participants' motivation can be maintained over a longer period. Future studies should collect this information to allow a more comprehensive assessment of training intensity and frequency.

## Conclusion

*MightyU* application, which combines sEMG with a specially designed gamified application, can be a motivating, valuable and low-cost tool for training at home among the special group of children with CP. Both, children and parents endorsed the device, and no adverse effects were reported. This indicates high feasibility and good usability. The initial results of the training are promising and suggest that repeated training improves overall performance in the game. Further investigations should be performed over a long study-period with a larger study group. It remains to be analyzed whether improved in-game-results correlate to improved motor skills or which functional changes to the Mighty-U application are required.

## Supporting information

**S1 Table. Children's questionnaire.**
(PDF)

**S2 Table. Parent's questionnaire.**
(PDF)

## Acknowledgments

The authors thank all participants and their families for supporting this study.

## Author contributions

**Conceptualization:** Lynn Eitner, Thomas Lücke.

**Data curation:** Lynn Eitner, Elnaz Farshadfar.

**Formal analysis:** Lynn Eitner, Lennart Lücke, Christoph Maier, Thomas Lücke.

**Funding acquisition:** Thomas Lücke.

**Investigation:** Lynn Eitner, Elnaz Farshadfar, Bettina Menzen.

**Methodology:** Lynn Eitner, Almut Weitkämper, Thomas Lücke.

**Project administration:** Thomas Lücke.

**Resources:** Lynn Eitner, Lennart Lücke, Elnaz Farshadfar, Christian Grüneberg, Christoph Maier, Almut Weitkämper, Bettina Menzen, Anja Burmann, Roman von Gehlen, Peter Konrad, Thomas Immich, Britta Karn, Patrick Filipowicz, Maximilian Pilk, Thomas Lücke.

**Software:** Anja Burmann, Roman von Gehlen, Peter Konrad, Thomas Immich, Britta Karn, Patrick Filipowicz, Maximilian Pilk.

**Supervision:** Lynn Eitner, Lennart Lücke, Elnaz Farshadfar, Christian Grüneberg, Christoph Maier, Almut Weitkämper, Bettina Menzen, Anja Burmann, Roman von Gehlen, Peter Konrad, Thomas Immich, Britta Karn, Patrick Filipowicz, Maximilian Pilk, Thomas Lücke.

**Validation:** Lynn Eitner, Lennart Lücke, Elnaz Farshadfar, Christian Grüneberg, Christoph Maier, Almut Weitkämper, Bettina Menzen, Anja Burmann, Roman von Gehlen, Peter Konrad, Thomas Immich, Britta Karn, Patrick Filipowicz, Maximilian Pilk, Thomas Lücke.

**Visualization:** Lynn Eitner, Lennart Lücke, Christoph Maier.

**Writing – original draft:** Lynn Eitner.

**Writing – review & editing:** Lynn Eitner, Lennart Lücke, Elnaz Farshadfar, Christian Grüneberg, Christoph Maier, Almut Weitkämper, Bettina Menzen, Anja Burmann, Roman von Gehlen, Peter Konrad, Thomas Immich, Britta Karn, Patrick Filipowicz, Maximilian Pilk, Thomas Lücke.

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
