## [Decision Letter · Decision Letter 0]

12 Mar 2025

Dear Dr. Eitner,

Thank you for submitting your manuscript to PLOS ONE. After careful consideration, we feel that it has merit but does not fully meet PLOS ONE’s publication criteria as it currently stands. Therefore, we invite you to submit a revised version of the manuscript that addresses the points raised during the review process.

**Please find the reviewers comment below, we would appreciate a detailed review and specific response to each point raised by the reviewers.**

We look forward to receiving your revised manuscript.

Kind regards,

Mshari Alghadier

Academic Editor

PLOS ONE

**Journal Requirements:**

1. When submitting your revision, we need you to address these additional requirements. Please ensure that your manuscript meets PLOS ONE's style requirements, including those for file naming. The PLOS ONE style templates can be found at https://journals.plos.org/plosone/s/file?id=wjVg/PLOSOne_formatting_sample_main_body.pdf and https://journals.plos.org/plosone/s/file?id=ba62/PLOSOne_formatting_sample_title_authors_affiliations.pdf 2. Thank you for stating in your Funding Statement: Support for the work was provided by Federal Ministry of Education and Research (Bundesministerium für Bildung und Forschung [BMBF], trial registration 13GW0299D). Please provide an amended statement that declares *all* the funding or sources of support (whether external or internal to your organization) received during this study, as detailed online in our guide for authors at http://journals.plos.org/plosone/s/submit-now.  Please also include the statement “There was no additional external funding received for this study.” in your updated Funding Statement. Please include your amended Funding Statement within your cover letter. We will change the online submission form on your behalf. 3. Please include captions for your Supporting Information files at the end of your manuscript, and update any in-text citations to match accordingly. Please see our Supporting Information guidelines for more information: http://journals.plos.org/plosone/s/supporting-information.

Reviewers' comments:

Reviewer's Responses to Questions

**Comments to the Author**

1. Is the manuscript technically sound, and do the data support the conclusions?

Reviewer #1: Yes

Reviewer #2: Yes

Reviewer #3: Partly

Reviewer #4: No

Reviewer #5: Yes

Reviewer #6: No

2. Has the statistical analysis been performed appropriately and rigorously?

Reviewer #1: Yes

Reviewer #2: Yes

Reviewer #3: No

Reviewer #4: No

Reviewer #5: Yes

Reviewer #6: No

3. Have the authors made all data underlying the findings in their manuscript fully available?

Reviewer #1: Yes

Reviewer #2: Yes

Reviewer #3: No

Reviewer #4: Yes

Reviewer #5: Yes

Reviewer #6: No

4. Is the manuscript presented in an intelligible fashion and written in standard English?

Reviewer #1: Yes

Reviewer #2: Yes

Reviewer #3: No

Reviewer #4: Yes

Reviewer #5: No

Reviewer #6: No

**Reviewer #1:** Thank you for your work and an interesting study. Gamification use in paediatric therapy is a really interesting and growing area, and this is interesting how it combines the value of biodfeedback too. well done. i am excited about the concept and the potential future application, though perhaps some edits are needed to strengthen this work (particularily for any rehabilitation therapists who would be reading this).

Abstract

- edit phrasing to be "children with CP"

- remove the word "only" before refused.

Introduction

- would it be worth adding some comment on this being a form of biofeedback (and what it known about biodfeedback)? Around line 63 it might be relevant to mention biofeedback here.

-line 89: should the phrasing be "prior to the start"

Methods

note that GMFCS levels use roman numerals (I-V)

How did you decide on which muscles to put the sensors on for each person (i.e. how did the physiotherapists decide on this)? Where the sensors placed across all of the muscle sites lists or were they targeted to the child? How did you ensure that the participants places the sensors on the correct locations at home? some of these muscles can be quite small, even more so on kids with CP- how do you avoid cross talk?

Results

line 211 "21 childen were contacted"

Do you have data on how long they used the system each time (minutes?), then also some sort of indication on the number of repititions they might have made in a particular movement/contraction? This would greatly strengthen this work.

Is there any option to also include some of the EMG or kinematic data within this manuscript? and how did it relate to the muscles that were needing to be targeted (e.g. in some cases it is not the muscle that we are aiming to recruit that is used to complete a task).

line 220: Is 'hand joint' meaning wrist joint? and is it ankle joint for feet? (knee extension and ankle dorsi flexion?)

You may want to check the formating of your tables so thaey are consistent (i.e. capital letters for Clinical data, Paresis etc).

It is hard to work out how many participants had one or more sensors on them, or what combination of muscles they had tested at time, could this be explained (maybe table 2 could have more explanation?)

line 234: 'episodes of dizziness"?

line 236: how did the parent report the muscle pain but not the child? Perhaps within this section it might be helpful to keep referring to the child- participants as children (not patients) and the parent (participants) as parents.

usability- removed the comma after 'Both'

Discussion

There is a flip back and forth between using CP and cerebral palsy.

line 306: extra space before the full stop

It feels like this is getting a little away from the application (and principles) of rehabilitation. Yes - specific task training is needed, and it generally applies to functional tasks based on the child’s therapy goals, that also need to active (not passive) – which is where the emg biofeedback approach is a positive step. "Engineer build training applications" could be misinterpreted here as being the solution.

Cost of other systems is mentioned as prohibitive, what would the approximate cost be of this system?

Can the game within the system be modified to suit children with higher GMFCS levels? These are the children who could likley benefit the most.

How does this approach minimise fatigue?

It might be important to comment that future work would be needed to evaluate if there are any functional changes with this tool.

**Reviewer #2:** Thank you for giving me the opportunity to review the manuscript - Mighty U – A portable sensor-based video game application for exercise training of children and adolescents with cerebral palsy.

Very interesting concept and very well-done study, though limited by the size of the sample. Never the less I hope it would pave way for larger studies. I recommend the publishing of the manuscript.

**Reviewer #3:** This study explores the effectiveness and feasibility of a virtual reality application called *MightyU*, designed for home-based neuro-rehabilitation of children with cerebral palsy (CP). The application uses an electromyographic (EMG) sensor to monitor the muscular movements and motor capabilities of the children, providing an interactive gaming experience that stimulates both fine and gross motor training.

ABSTRACT

Lines 8-9: Could you clarify what you mean by "kinematic electromyography"? Are you referring exclusively to surface EMG?

Line 24: Please provide the SUS score.

INTRODUCTION

Line 30: Replace "Infantile Cerebral Palsy" with "Cerebral Palsy (CP)".

Line 31: Replace "affected individuals" with "individuals with CP".

Line 34: Replace "affected children" with "children with CP".

Line 54: Replace "cerebral palsy" with "CP".

Line 60: Add a space before reference (10).

Line 61: Replace "sensor-based surface EMG technology" with "surface EMG".

Line 67: Replace "underscore" with "underline" (?)

Line 68: Replace "CP patients" with "children with CP".

Lines 73-78: Rephrase and avoid the numbered list. Try to combine the points into a coherent sentence.

Line 78: Clarify the studies referenced (14-17).

METHODS

Line 82: Replace "subjects" with "participants".

Line 87: Italicize "MightyU" and remove the quotation marks.

Lines 93-94: Remove the brackets around "information technology".

Line 95: Spell out the acronym "IT" as "information technology".

Line 100: Provide a more detailed description of the movements and which muscles were targeted.

Line 101: Specify the anatomical points where sensors were placed.

Line 107: Replace "cerebral palsy" with "CP".

Line 108: Remove the phrase "a recent work by".

Line 110: Clarify how the surveys were distributed and structured.

Line 117: Remove the word "special".

Line 118: Specify the type of muscles or muscle groups targeted.

Line 122: Clarify what you mean by "degree of muscle tension"—is it referring to the amplitude of the EMG signal?

Line 123: Explain the calibration procedure or provide a reference for it.

Line 129: Clarify what criteria were used to add an obstacle and how many more obstacles were quantified.

Line 132: Specify which muscles were involved and explain how you determined if the maximum contraction was performed by the child.

Line 142: Specify all the data collected in the training history (e.g., number of sessions, duration, etc.).

Line 147: Add references for the questionnaires used.

Line 161: Cite the supplementary materials.

Line 174: Move the "Subjects" paragraph (renaming it "Participants") before the section "MightyU Game Application".

Line 178: Replace "patients" with "children".

Line 182: Clarify GMFCS Levels 1-4 and explain how they were used in the study.

Line 187: Clarify the exclusion criteria.

Line 189: Rename the paragraph to "Timing of Acquisition" and remove the reference to "measurement procedure".

Lines 201-208: Specify which types of variables were analyzed and clarify which variables were measured in the study.

RESULTS

Line 210: Delete "subjects" and use "participants".

Line 238: Rephrase "N = 3 children" as "3 children".

Lines 249-254: The quartiles are not indicated in the results.

Line 269: Delete "(N=1)"—it is clear in the text.

Line 279: Rephrase "independent of the starting level" to "independently of the starting level".

DISCUSSION

Line 293: Clarify the use of "surface EMG sensor".

Line 294: Use "muscle activation" instead of "muscle contraction".

Avoid numbered lists; try to rephrase the points in narrative form.

Line 313: Delete "N=" and replace "subjects" with "individuals".

Line 336: Rephrase "sensor-based surface EMG (sEMG) technology".

Line 347: Delete "N=".

Lines 356-357: "Allowed the intensity of muscle contraction to be quantified"—How was this quantified? Please clarify the methodology.

Line 363: Replace "affected" with "children with CP".

Line 366: Remove the paragraph name "Interpretation".

Line 386: Replace "level of disability" with "classification level".

Explain the limitations of the study and the potential for future developments more clearly.

CONCLUSION

Avoid using "patients"; replace with "children" or a suitable synonym.

TABLES

Table 2: Use the same font throughout the table.

Table 1 Caption: "Participants' Clinical Characteristics".

Refer to Table 2 in the text when mentioning "muscles".

FIGURES

Figure 2: Add the statistical analysis performed to the figure.

Figure 4: Replace "GMFC score" with "GMFCS level" and add a legend explaining the colors.

GENERAL CONSIDERATIONS

Revise the "Methods" section to use a more scientific tone.

Provide the results with the respective statistical analyses.

Clarify the type of variables used to control the game—only EMG amplitude?

Some points could benefit from rephrasing to improve clarity.

Add a photo of the experimental set up.

**Reviewer #4:** Lynn Eitner and the group took the most trending virtual reality approach to develop a gamming concept as a therapeutic or rehabilitatory tool for cerebral palsy patients. To edge their virtual gaming competitive market, the authors combined sensor-based surface EMG technology as the human operating control for the video game, which can detect and record small muscle contractions over time. This gave the authors to tailor the game specifically for patient needs or disability, targeting muscle of interest which made the tool versatile. I really appreciate their thought process, and the authors conceptually contributed towards the merit of personalized therapeutic research.

Here I would like to address a few important areas of my concern and highlighting the scope of improvement for the authors.

Overall concerns:

1.The authors designed their manuscript around the virtual reality game, its development, tailoring to the patient needs, surveys, etc, but the aim of the article suggests ‘… exercise training of the children and adolescents with cerebral palsy’. They failed to mention any information regarding how their device or game improved the scientific or theoretical knowledge of treatment or therapeutic/rehabilitatory observations of their patient under study.

2.The game includes the sensors of EMG, that could have been their main focused since it was the bridge between virtual game development and scientific advancement in neuro-therapeutic field. There was no data provided in the current form of this article in this regard. In the discussion section authors promised to provide that in their future publications. I think it would be good to combine their present and future goals together to establish a story with good scientific merit.

Concerns related to the current form of the article:

1.In the introduction, authors mentioned modification in the MightyU project (Page 5 line 69): Is the program upgradable? Meaning, is it possible to modify or adjust the difficulty level of the game in accordance with patients’ impairments and/or improvement? More justification would be appreciated for the readers.

2.Method section (page 6 line 80) suggests that the project has been approved by the ethics committee in 2020. The obvious question comes, why has there not been any follow up study conducted?

3.The game development section (Page 7 line 113) described the game. What is unique about the logic/algorithm of the game which makes this development unique and tailored to the patient as compared to their competitors’ product? Little more description would be helpful for the reader to understand the rational.

4.The section also talks about the use of EMG in the playing process and gaming score. The patient needs to contract or relax their muscles to keep the virtual player floating on the screen during the game. Can authors comment on whether the recording from EMG data pattern and the gaming score comparable? It would be helpful to directly relate their gamification process with therapeutic progress.

5.Measurement procedure (Page 10 line 190): Was the EMG data collected during the training, practice or trail sessions being used in medical research in any form? Was the data evaluated by the patient’s therapist? Detailed description of whether or how that data has been processed will give the article a justification for developing muscle targeted gamming device for CP patients.

6.Figure 3 required more explanation why the GEQ rating of healthy parents and children with CP are almost identical in terms of competence and challenge parameters.

7.Figure 4: The comparative line graphs show almost 70% cases where the GMFC score improved from 1st to 2nd assessment. Based on the gaming logic explained in this article, neurologically that graph implies strengthening of muscle or improvement on controlling muscle contraction/relaxation. These are very important parameters for CP therapeutic aspect. The question raises why this result has not been correlated with EMG data collected during the assessments?

**Reviewer #5:**  Thank you for this submission. This is an interesting study and the MightyU seems like a valuable addition to a home exercise program.

*While the study appears to be sound, the language is sometimes unclear, making it difficult to follow. I advise the authors work with a writing coach or copyeditor to improve the flow and readability of the text.

*I would recommend no beginning sentences with "N=XX" and use percentages and not 2/3 (line 304) when appropriate. This was noted only toward the end of the paper.

*The description for the GMFCS level 4 participants in Figure 4 is listed as purple and violet. It would be best to pick on color descriptor (purple).

*I would not call the MightyU a VR program. It is a game, but not virtual reality based on your description.

*It would be interesting to see if the participants demonstrated increased contraction strength as a result of their use of the device. Is this information available or will this be a future study?

*How often should this be re-calibrated during their use of the product? If I understand correctly, the device is calibrated to the participant once when treatment is initiated and then was not re-calibrated during the one-week trial.

**Reviewer #6:** Overall summary

This non-randomised, non-controlled experimental study measures the feasibility, acceptance, and intrinsic effectiveness (improvement of the same performance which is trained) of a novel rehabilitation videogame. The main strength of the study is the type of sensorisation which is employed, i.e. surface EMG. In fact, most human-machine interfaces are based on kinematic signals provided by motion-capture systems, whereas electromyography is seldom used. Another strong point is that different, disorder-specific muscles were targeted on the basis of physiotherapists' advice. Major weaknesses are: the game is very simple and looks rather repetitive, no measures of clinical effectiveness were taken, no control sample was recruited, the clinical sample is small and very diverse as far as functional impairment, topography of motor disorder, and intellectual disabilities are concerned. For these reasons, I consider it of poor scientific quality, though original and interesting for rehabilitation practice. I am thus in favour of its rejection.

Answers to review questions

1) The manuscript isn't very sound because it doesn't fully decribe nor explain the experimental setup. Moreover, it doesn't include any clinical outcome measure nor controls of any sort. No hypotheses are made and therefore data can neither support nor reject the hypotheses.

2) Statistical analysis is virtually absent, since data are only descriptive. Some analysis has been carried out, but no conclusions can be drawn.

3) I couldn't find the full clinical description of cases, nor the association between clinical pictures (type, distribution of motor disorders) and experimental conditions.

4) The manuscript is written in standard English, but it hasn't been revised.

Details by section

Introduction

Page 5, line 65. Please explicit the advantages of sEMG compared with other sensors for exergames.

Page 5, line 78. Please add the expected results of your study.

Methods

Page 7, line 117. What is special about the sensors?

Page 7, line 120. Setup specifications are not clear enough. Please add technical data such as latency and measurement errors. Please specify how many sensors were used and in which configuration. Cross-talk issues? Latency issues? Was the bluetooth connection for each sensor or were sensors cabled to a single transmitter?

By the way, does each child use always the same muscle(s) for gearing the game? Does each child use one or more muscles at a time? Is game duration always the same or does it change according to levels? Procedure requires clarification.

Page 7, line 122. Define 'muscle tension': which signal measure was used and how? Error? Page 7, line 123. Explain the calibration procedure.

Page 9, line 170. How did you normalise the measure 'number of coins' across different levels, game durations (if variable), and muscles tested? Why didn't you investigate any clinical outcome measure, such as muscle strength, endurance, gross-motor function, attention, QoL?

Results

Page 12, line 220. Which 'hand joint' do you mean exactly, the radio-carpal joint? What does 'stretching their feet' mean, maybe plantar flexion? What CP type (topography and type of motor disorder) was matched with what muscles? Clinical characterisation is very poor.

Discussion

I expected to gain more insight into the pros and contras of this particular methodology in comparison with other 1) types of exergame, 2) types of sensorisation, 3) training methods. I haven't found much of the sort.

Page 20, line 336. Unfinished sentence.

**Do you want your identity to be public for this peer review?** For information about this choice, including consent withdrawal, please see our Privacy Policy

Reviewer #1: No

Reviewer #2: No

Reviewer #3: No

Reviewer #4: **Yes:** Srikanya Kundu

Reviewer #5: No

Reviewer #6: No

---

## [Author Response · Author response to Decision Letter 1]

15 Sep 2025

(Content below is also included in cover letter attached to revised manuscript)

Reply to reviewers

Reviewer #1:

Introduction

Would it be worth adding some comment on this being a form of biofeedback (and what it known about biodfeedback)? Around line 63 it might be relevant to mention biofeedback here.

Author’s reply:

Thank you for the idea of making the very useful connection to biofeedback. We will now explain this method, provide an insight into how biofeedback is used in the neurorehabilitation of children, and establish a connection to our training device. (page 5, lines 65-71)

Methods

How did you decide on which muscles to put the sensors on for each person (i.e. how did the physiotherapists decide on this)? Where the sensors placed across all of the muscle sites lists or were they targeted to the child? How did you ensure that the participants places the sensors on the correct locations at home? some of these muscles can be quite small, even more so on kids with CP- how do you avoid cross talk?

It is hard to work out how many participants had one or more sensors on them, or what combination of muscles they had tested at time, could this be explained (maybe table 2 could have more explanation?)

Author’s reply:

This feedback is also very helpful. We have now added a subsection entitled "Muscle selection and sEMG sensor placement" to the methods section to describe these points in more detail. (page 10)

Results

Do you have data on how long they used the system each time (minutes?), then also some sort of indication on the number of repetitions they might have made in a particular movement/contraction? This would greatly strengthen this work.

Author’s reply:

We added how many days the children used the system for training at home. Unfortunately, exact minute details cannot currently be extracted.

Is there any option to also include some of the EMG or kinematic data within this manuscript? and how did it relate to the muscles that were needing to be targeted (e.g. in some cases it is not the muscle that we are aiming to recruit that is used to complete a task).

Author’s reply:

Three reviewers suggest including sEMG data in this manuscript (see below). Our original plan was to focus solely on the feasibility and usability of the new video game application in this publication. The motion analyses, which, in addition to sEMG data, also include kinematic data from marker-based videogrammetry (2D) and a measurement using an inertial sensor system (IMU) (3D), were intended to be the subject of a subsequent publication. We have now decided to include the topic of sEMG in this publication as an example (see figure 5).

Discussion

It feels like this is getting a little away from the application (and principles) of rehabilitation. Yes - specific task training is needed, and it generally applies to functional tasks based on the child’s therapy goals, that also need to active (not passive) – which is where the emg biofeedback approach is a positive step. "Engineer build training applications" could be misinterpreted here as being the solution.

Author’s reply:

Thank you for this important comment. We certainly don't want the video game to be misunderstood as a replacement for all conventional and proven therapy options. It is simply intended to be a child-friendly and motivating addition to everyday therapy. We have clarified the relevant passages.

Can the game within the system be modified to suit children with higher GMFCS levels? These are the children who could likely benefit the most.

Author’s reply:

This option is desirable for further development of the system. For the initial test runs, we needed a system that is as broadly applicable as possible, so that the largest possible patient group can try out the application and adequately evaluate it.

It might be important to comment that future work would be needed to evaluate if there are any functional changes with this tool.

Author’s reply:

Thank you for this idea, we added this hint to the conclusions.

Reviewer #3:

ABSTRACT

Lines 8-9: Could you clarify what you mean by "kinematic electromyography"? Are you referring exclusively to surface EMG?

Author’s reply:

We agree with the reviewer that the term is misleading. Therefore, we have removed it from the text passage and defined it as follows: “A surface electromyography sensor detects muscle activation, which is translated into in-game actions. In this way, targeted muscle activity is used to collect coins during gameplay.” (page 2, lines 8ff)

METHODS

Provide a more detailed description of the movements and which muscles were targeted. Specify the anatomical points where sensors were placed. Specify the type of muscles or muscle groups targeted.

Line 132: Specify which muscles were involved and explain how you determined if the maximum contraction was performed by the child.

Author’s reply:

Thank you for feedback. We added in the section methods additional information concerning selected muscles and positions sensors applied and the target movement in table 2.

Explain the calibration procedure or provide a reference for it.

Author’s reply:

We appreciate your input and enhance the explanation in the method section with additional definition on sensor calibration procedure for the individual patients. (page 10, lines 169ff)

Clarify what criteria were used to add an obstacle and how many more obstacles were quantified.

Author’s reply:

Thanks for your raising this topic. We added additional information on the same standardized level all included patients played during the study visits. This level did not have any obstacles included. (page 11, lines 184ff)

Specify which types of variables were analyzed and clarify which variables were measured in the study.

Author’s reply:

Thanks for your feedback and input. In this first manuscript we are only reviewing the feasibility and usability of the new training assessment. As measuring tool a survey has been used among the children and parents. We now explain the statistical procedures in more detail in the results section. (pages 15ff)

DISCUSSION

Lines 356-357: "Allowed the intensity of muscle contraction to be quantified"—How was this quantified? Please clarify the methodology.

Author’s reply:

Thanks for your comment. The intensity of contraction can be monitored during the gameplay by reviewing the movement of the character in the game. Compare explanations on biofeedback in the introduction. (page 5, lines 62ff and page 9, lines 148ff)

Explain the limitations of the study and the potential for future developments more clearly.

Author’s reply:

We appreciate your feedback. Limitations, improvement need and further development potentials in different areas have been stated in the discussion section.

FIGURES

Figure 2: Add the statistical analysis performed to the figure.

Author’s reply:

Thanks for your feedback, we have added percentages of the positive and negative answers to get an overview about the frequency. Explanations to the figures are indicated under Feasibility. (see figure 3)

GENERAL CONSIDERATIONS

Revise the "Methods" section to use a more scientific tone. Some points could benefit from rephrasing to improve clarity.

Author’s reply:

Several sections have been updated accordingly.

Provide the results with the respective statistical analyses.

Author’s reply:

Statistical analyses have been added.

Clarify the type of variables used to control the game—only EMG amplitude?

Author’s reply:

This is correct, only EMG sensor has been used.

Add a photo of the experimental set up.

Author’s reply:

We added a photo of the EEG-sensor located on a child’s forearm. (see Fig.1)

Reviewer #4:

1.The authors failed to mention any information regarding how their device or game improved the scientific or theoretical knowledge of treatment or therapeutic/rehabilitatory observations of their patient under study.

Author’s reply:

We can agree with this objection. The patients only tested the training tool at home for one week, and the initial aim of this intervention was to test its usability and feasibility. The next step is to assess its therapeutic benefit. However, we are, of course, aware that the intervention phase will need to last significantly longer. It is part of a further study.

2.The game includes the sensors of EMG, that could have been their main focused since it was the bridge between virtual game development and scientific advancement in neuro-therapeutic field. There was no data provided in the current form of this article in this regard. In the discussion section authors promised to provide that in their future publications. I think it would be good to combine their present and future goals together to establish a story with good scientific merit.

Author’s reply:

Same topic raised by other reviewer 1. Compare above.

Concerns related to the current form of the article:

1. In the introduction, authors mentioned modification in the MightyU project (Page 5 line 69): Is the program upgradable? Meaning, is it possible to modify or adjust the difficulty level of the game in accordance with patients’ impairments and/or improvement? More justification would be appreciated for the readers.

Author’s reply:

Same topic raised by other reviewer. Compare above

2. Method section (page 6 line 80) suggests that the project has been approved by the ethics committee in 2020. The obvious question comes, why has there not been any follow up study conducted? --?! :D

Author’s reply:

Thanks for your feedback. There has been a lot of interest in a follow up study from various parties. Planning activities are ongoing.

3. The game development section (Page 7 line 113) described the game. What is unique about the logic/algorithm of the game which makes this development unique and tailored to the patient as compared to their competitors’ product? Little more description would be helpful for the reader to understand the rational.

Author’s reply:

Thank you for this comment. We added this information in the introduction. (page 5, lines 61ff)

4. The section also talks about the use of EMG in the playing process and gaming score. The patient needs to contract or relax their muscles to keep the virtual player floating on the screen during the game. Can authors comment on whether the recording from EMG data pattern and the gaming score comparable? It would be helpful to directly relate their gamification process with therapeutic progress.

Author’s reply:

We have included an example to illustrate the EMG analysis, which several reviewers requested. See Figure 6 and comments page 20, lines 341ff.

5. Measurement procedure (Page 10 line 190): Was the EMG data collected during the training, practice or trail sessions being used in medical research in any form? Was the data evaluated by the patient’s therapist? Detailed description of whether or how that data has been processed will give the article a justification for developing muscle targeted gaming device for CP patients. (line 205)

Author’s reply:

See question above. The next step is to assess the advices therapeutic benefit.

6.Figure 3 required more explanation why the GEQ rating of healthy parents and children with CP are almost identical in terms of competence and challenge parameters.

Author’s reply:

Thank you for your feedback. Further explanations have been added to the manuscript. Parents' responses are based on observations while their children were using the game application. (page 17, line 293)

7.Figure 4: The comparative line graphs show almost 70% cases where the GMFC score improved from 1st to 2nd assessment. Based on the gaming logic explained in this article, neurologically that graph implies strengthening of muscle or improvement on controlling muscle contraction/relaxation. These are very important parameters for CP therapeutic aspect. The question raises why this result has not been correlated with EMG data collected during the assessments?

Author’s reply:

Three reviewers suggest including sEMG data in this manuscript (see above). Our original plan was to focus solely on the feasibility and usability of the new video game application in this publication. The motion analyses, which, in addition to sEMG data, also include kinematic data from marker-based videogrammetry (2D) and a measurement using an inertial sensor system (IMU) (3D), were intended to be the subject of a subsequent publication. We have now decided to include the topic of sEMG in this publication as an example (see figure 6).

Reviewer #5:

Thank you for this submission. This is an interesting study and the MightyU seems like a valuable addition to a home exercise program.

*While the study appears to be sound, the language is sometimes unclear, making it difficult to follow. I advise the authors work with a writing coach or copyeditor to improve the flow and readability of the text.

Author’s reply:

We appreciate your feedback. Several passages of the manuscript have been updated to simplify and be more sound on the argumentation structure.

*I would not call the MightyU a VR program. It is a game, but not virtual reality based on your description.

Author’s reply:

Thanks for the very valuable comment. It has been corrected.

*It would be interesting to see if the participants demonstrated increased contraction strength as a result of their use of the device. Is this information available or will this be a future study?

Author’s reply:

Children only used the Game for a rather short duration of one week, therefore we did not intend to review improvement contraction strength. For a future study this investigation will be in scope.

*How often should this be re-calibrated during their use of the product? If I understand correctly, the device is calibrated to the participant once when treatment is initiated and then was not re-calibrated during the one-week trial.

Author’s reply:

Thanks for your feedback. Additional information on the calibration procedure has been added to clarify this point. (page 10, lines 170ff)

Reviewer #6: Overall summary

1) The manuscript isn't very sound because it doesn't fully decribe nor explain the experimental setup. Moreover, it doesn't include any clinical outcome measure nor controls of any sort. No hypotheses are made and therefore data can neither support nor reject the hypotheses.

Author’s reply:

We thank you for your review of the manuscript. The methodology was significantly expanded and clarified during the revision. The initial evaluation focused on assessing the feasibility and usability of the novel game application. For this purpose, questionnaires were conducted with children and their observing parents. Therefore, the assessment of the actual target group—children with cerebral palsy—was of interest, so no control group was planned in this first step.

2) Statistical analysis is virtually absent, since data are only descriptive. Some analysis has been carried out, but no conclusions can be drawn.

Author’s reply:

On this point, we do not entirely agree with the reviewer. Of course, with a sample size of 21 children (17 fully completed questionnaires), a comprehensive statistical analysis is not possible, but a descriptive evaluation. Nevertheless, this is a common method for generating results that also allow first conclusions and summaries.

3) I couldn't find the full clinical description of cases, nor the association between clinical pictures (type, distribution of motor disorders) and experimental conditions.

What CP type (topography and type of motor disorder) was matched with what muscles? Clinical characterisation is very poor.

Author’s reply:

We thank you for rising this issue. Clinical characterizations of the patient group have been added (Table 1), and the methods now explain in more detail the criteria used to select the target movements and muscle groups.

4) The manuscript is written in standard English, but it hasn't been revised.

Author’s reply:

Thank you for your feedback. Many passages has been revised.

Methods

Page 7, line 117. What is special about the sensors?

Author’s reply:

We added information about the sensors under the section “the

---

## [Decision Letter · Decision Letter 1]

5 Oct 2025

Dear Dr. Eitner,

Thank you for submitting your manuscript to PLOS ONE. After careful consideration, we feel that it has merit but does not fully meet PLOS ONE’s publication criteria as it currently stands. Therefore, we invite you to submit a revised version of the manuscript that addresses the points raised during the review process.

We look forward to receiving your revised manuscript.

Kind regards,

Mshari Alghadier

Academic Editor

PLOS ONE

Journal Requirements:

Reviewers' comments:

Reviewer's Responses to Questions

**Comments to the Author**

Reviewer #3: (No Response)

Reviewer #4: All comments have been addressed

Reviewer #5: All comments have been addressed

2. Is the manuscript technically sound, and do the data support the conclusions?

Reviewer #3: Yes

Reviewer #4: Yes

Reviewer #5: Yes

3. Has the statistical analysis been performed appropriately and rigorously?

Reviewer #3: N/A

Reviewer #4: Yes

Reviewer #5: Yes

4. Have the authors made all data underlying the findings in their manuscript fully available?

Reviewer #3: Yes

Reviewer #4: Yes

Reviewer #5: Yes

5. Is the manuscript presented in an intelligible fashion and written in standard English?

Reviewer #3: Yes

Reviewer #4: Yes

Reviewer #5: Yes

Reviewer #3: General comments

The manuscript presents an interesting study on the use of a gamified sEMG-based application for home-based rehabilitation of children with cerebral palsy (CP). The work is well-structured, and the preliminary data are promising. However, several aspects require attention before submission:

Abstract

break long sentences, clarify the effect size (68% improvement).

Include the number of participants in the Methods section of the abstract.

Terminology and consistency

Muscles:

quadrizeps femoris → quadriceps femoris

digitorum profundus palmarflexion → flexor digitorum profundus – wrist/finger flexion

peroneus longus abduction → peroneus longus – foot eversion

Correlation coefficient: replace “onkoeffic ient”.

Standardize abbreviations: sEMG, EMG.

Methods

Break long sentences in the EMG sensor and gameplay descriptions.

Use bullet points for technical parameters (A/D conversion, filtering, RMS).

Clarify access to data on the web portal.

Clearly describe the GEQ and SUS scales.

line 223: remove ) at the end of the sentence

Results

Correct muscle names and movements in Table 2.

Standardize data presentation across text, tables, and figures (percentages vs. medians).

Check discrepancies between the number of participants reported in text and figures.

Discussion

Some sections are redundant (e.g., paragraphs 356–398).

Avoid repetition of previously reported results.

line 389 avoid to use patients. Children should be better.

Explain the limitations of the study and the potential for future developments more clearly.

Conclusions

Simplify long sentences.

Summarize in 2–3 concise sentences: safety, usability, motivation, potential for home-based training.

Tables and figures

Table 2: correct muscle names and movements.

Figures 1–6: check readability and legends; standardize numbering.

GEQ/SUS: standardize style and units.

Table 3: replace a comma with . 0,32 -> 0.32

replace correlationkoefficent with "correlation coefficient"

Figure 5: add "S" to "GMFC"

Style and language

Shorten long sentences and break overly complex periods.

Standardize verb tense: past for methods, present for results and interpretation.

Check punctuation, spacing, and terminological consistency.

Conclusion

The manuscript has high scientific potential. After correcting terminology, standardizing data, reducing redundancy, and improving style, it will be suitable for pubblicaiton.

TABLE 2

I would correct the table as follows.

Biceps brachii Flexion of the forearm

Brachioradialis Flexion of the forearm / assists supination

Flexor digitorum profundus Flexion of the fingers

Extensor carpi radialis longus Extension of the wrist

Extensor digitorum Extension of the fingers

Tibialis anterior Dorsiflexion of the foot

Quadriceps femoris Extension of the knee joint

Peroneus longus Eversion of the foot

Reviewer #4: All comments were address to the best of their ability and available data. The current manuscript has been considered as a first part of their ongoing reserach. Looking forward to their upcoming publication.

Reviewer #5: Thank you for the opportunity to review this resubmission. A few minor comments. there is a typo in Table 3 Correlation Coefficient.

Figure 5, left axis of the figure is hard to read, recommend labeling every 5 coins.

This is a much better submission than the original. Thank you for addressing our concerns.

**Do you want your identity to be public for this peer review?** For information about this choice, including consent withdrawal, please see our Privacy Policy

Reviewer #3: No

Reviewer #4: No

Reviewer #5: No

---

## [Author Response · Author response to Decision Letter 2]

25 Nov 2025

Reply to reviewers

Thank you very much for your support, your ideas that have greatly enriched the manuscript, and the time you have taken to edit it.

Reviewer #3:

Abstract

break long sentences, clarify the effect size (68% improvement).

Include the number of participants in the Methods section of the abstract.

Author’s reply:

Thank you very much for these comments. We have revised the text passages and shortened the sentences.

Terminology and consistency

Muscles:

quadrizeps femoris → quadriceps femoris

digitorum profundus palmarflexion → flexor digitorum profundus – wrist/finger flexion

peroneus longus abduction → peroneus longus – foot eversion

Correlation coefficient: replace “onkoeffic ient”.

Standardize abbreviations: sEMG, EMG.

Author’s reply:

Thank you very much for the suggestions. We have adopted them all.

Methods

Break long sentences in the EMG sensor and gameplay descriptions.

Clarify access to data on the web portal.

line 223: remove) at the end of the sentence

Author’s reply:

Thank you very much for these comments. We shortened the sentences (page 9, lines 153ff) and added a description of data assessment on the web portal (page 10, lines 198ff).

Use bullet points for technical parameters (A/D conversion, filtering, RMS).

Author’s reply:

This is a good suggestion. We tried bullet points but found that reducing the descriptions greatly limited comprehensibility. For this reason, we have retained complete sentences.

Clearly describe the GEQ and SUS scales.

Author’s reply:

We have added a reference to the material in the supplements. This contains all the questions from both questionnaires.

Results

Correct muscle names and movements in Table 2.

Author’s reply:

Thank you very much. We have implemented this suggestion and completely revised Table 2.

Standardize data presentation across text, tables, and figures (percentages vs. medians).

Author’s reply:

We have checked all texts, tables, and illustrations for discrepancies and have not found any inconsistencies.

Check discrepancies between the number of participants reported in text and figures.

Author’s reply:

All evaluations refer to the 16 participants who took part in both visits. Therefore, all tables and figures include 16 children. Nevertheless, it was relevant for the results of the study to also mention the others, those who did not want to participate and those who only took part in the first visit. We describe this in the text on page 14, lines 257 ff.

Discussion

Some sections are redundant (e.g., paragraphs 356–398).

Avoid repetition of previously reported results.

Author’s reply:

We appreciate this comment, but we believe that it is usual to summarize the results again at the beginning of the discussion for the sake of readability and to provide a quick overview. We think that it is helpful for the flow of reading. For this reason, we have decided not to delete the passages.

line 389 avoid to use patients. Children should be better.

Explain the limitations of the study and the potential for future developments more clearly.

Author’s reply:

We have incorporated these comments and added some explanatory sentences to the discussion (page 24, lines 473 ff).

Conclusions

Simplify long sentences.

Summarize in 2–3 concise sentences: safety, usability, motivation, potential for home-based training.

Author’s reply:

We shortened the conclusion a bit and removed long sentences.

Tables and figures

Table 2: correct muscle names and movements.

Figures 1–6: check readability and legends; standardize numbering.

GEQ/SUS: standardize style and units.

Table 3: replace a comma with . 0,32 -> 0.32

replace correlationkoefficent with "correlation coefficient"

Figure 5: add "S" to "GMFC"

Author’s reply:

This feedback was very helpful. We have implemented all points in the tables and figures.

Standardize verb tense: past for methods, present for results and interpretation. Check punctuation, spacing, and terminological consistency.

Author’s reply:

Thank you very much for this tip. We have made the necessary corrections.

TABLE 2

I would correct the table as follows.

Biceps brachii Flexion of the forearm

Brachioradialis Flexion of the forearm / assists supination

Flexor digitorum profundus Flexion of the fingers

Extensor carpi radialis longus Extension of the wrist

Extensor digitorum Extension of the fingers

Tibialis anterior Dorsiflexion of the foot

Quadriceps femoris Extension of the knee joint

Peroneus longus Eversion of the foot

Author’s reply:

We appreciate this excellent suggestion and have incorporated everything into the table exactly as you suggested.

Reviewer #4:

All comments were address to the best of their ability and available data. The current manuscript has been considered as a first part of their ongoing reserach. Looking forward to their upcoming publication.

Author’s reply:

We would like to thank you for your support.

Reviewer #5:

Thank you for the opportunity to review this resubmission. A few minor comments. there is a typo in Table 3 Correlation Coefficient.

Figure 5, left axis of the figure is hard to read, recommend labeling every 5 coins.

This is a much better submission than the original. Thank you for addressing our concerns.

Author’s reply:

Thank you very much for pointing this out. We have corrected it accordingly.

---

## [Decision Letter · Decision Letter 2]

10 Dec 2025

MightyU – A portable sensor-based video game application for exercise training of children and adolescents with cerebral palsy

PONE-D-24-54773R2

Dear Dr. Eitner,

We’re pleased to inform you that your manuscript has been judged scientifically suitable for publication and will be formally accepted for publication once it meets all outstanding technical requirements.

Kind regards,

Mshari Alghadier

Academic Editor

PLOS One

Additional Editor Comments (optional):

Reviewers' comments:

Reviewer's Responses to Questions

**Comments to the Author**

Reviewer #3: All comments have been addressed

Reviewer #5: All comments have been addressed

2. Is the manuscript technically sound, and do the data support the conclusions?

Reviewer #3: Yes

Reviewer #5: Yes

3. Has the statistical analysis been performed appropriately and rigorously?

Reviewer #3: N/A

Reviewer #5: Yes

4. Have the authors made all data underlying the findings in their manuscript fully available?

Reviewer #3: No

Reviewer #5: Yes

5. Is the manuscript presented in an intelligible fashion and written in standard English?

Reviewer #3: Yes

Reviewer #5: Yes

Reviewer #3: The review has been completed and I have no further comments. The document is clear, comprehensive, and consistent. It can be accepted.

Reviewer #5: Thank you for addressing the concerns listed in the previous review. I have no further concerns about this submission.

**Do you want your identity to be public for this peer review?** For information about this choice, including consent withdrawal, please see our Privacy Policy

Reviewer #3: No

Reviewer #5: No

---

## [Editor Report · Acceptance letter]

PONE-D-24-54773R2

PLOS One

Dear Dr. Eitner,

I'm pleased to inform you that your manuscript has been deemed suitable for publication in PLOS One. Congratulations! Your manuscript is now being handed over to our production team.

Kind regards,

on behalf of

Dr. Mshari Alghadier

Academic Editor

PLOS One